# Design, Synthesis, and Comparison of PLA-PEG-PLA and PEG-PLA-PEG Copolymers for Curcumin Delivery to Cancer Cells

**DOI:** 10.3390/polym15143133

**Published:** 2023-07-23

**Authors:** Neda Rostami, Farzaneh Faridghiasi, Aida Ghebleh, Hadi Noei, Meisam Samadzadeh, Mohammad Mahmoudi Gomari, Alireza Tajiki, Majid Abdouss, Alireza Aminoroaya, Manisha Kumari, Reza Heidari, Vladimir N. Uversky, Bryan R. Smith

**Affiliations:** 1Department of Chemistry, Amirkabir University of Technology, Tehran 1591634311, Iran; 2Department of Tissue Engineering and Regenerative Medicine, Faculty of Advanced Technologies in Medicine, Iran University of Medical Sciences, Tehran 1449614535, Iran; 3School of Advanced Technologies in Medicine, Isfahan University of Medical Sciences, Isfahan 8174673461, Iran; 4Department of Medical Biology and Genetics, Faculty of Medicine, Istinye University, Istanbul 34010, Turkey; 5Department of Molecular Biology and Genetics, Faculty of Engineering and Natural Sciences, Istinye University, Istanbul 34010, Turkey; 6Department of Medical Biotechnology, Faculty of Allied Medicine, Iran University of Medical Sciences, Tehran 1449614535, Iran; 7Department of Biomedical Engineering and Institute for Quantitative Health Science and Engineering, Michigan State University, East Lansing, MI 48824, USA; 8Research Center for Cancer Screening and Epidemiology, AJA University of Medical Sciences, Tehran 1411718541, Iran; 9Department of Molecular Medicine and USF Health Byrd Alzheimer’s Research Institute, Morsani College of Medicine, University of South Florida, Tampa, FL 33612, USA

**Keywords:** breast cancer, curcumin, copolymer, drug delivery, nano-informatics, biomaterials, PEG, PLA

## Abstract

Curcumin (CUR) has potent anticancer activities, and its bioformulations, including biodegradable polymers, are increasingly able to improve CUR’s solubility, stability, and delivery to cancer cells. In this study, copolymers comprising poly (L-lactide)-poly (ethylene glycol)-poly (L-lactide) (PLA-PEG-PLA) and poly (ethylene glycol)-poly (L-lactide)-poly (ethylene glycol) (PEG-PLA-PEG) were designed and synthesized to assess and compare their CUR-delivery capacity and inhibitory potency on MCF-7 breast cancer cells. Molecular dynamics simulations and free energy analysis indicated that PLA-PEG-PLA has a higher propensity to interact with the cell membrane and more negative free energy, suggesting it is the better carrier for cell membrane penetration. To characterize the copolymer synthesis, Fourier transform-infrared (FT-IR) and proton nuclear magnetic resonance (^1^H-NMR) were employed, copolymer size was measured using dynamic light scattering (DLS), and their surface charge was determined by zeta potential analysis. Characterization indicated that the ring-opening polymerization (ROP) reaction was optimal for synthesizing high-quality polymers. Microspheres comprising the copolymers were then synthesized successfully. Of the two formulations, PLA-PEG-PLA experimentally exhibited better results, with an initial burst release of 17.5%, followed by a slow, constant release of the encapsulated drug up to 80%. PLA-PEG-PLA-CUR showed a significant increase in cell death in MCF-7 cancer cells (IC_50_ = 23.01 ± 0.85 µM) based on the MTT assay. These data were consistent with gene expression studies of Bax, Bcl2, and hTERT, which showed that PLA-PEG-PLA-CUR induced apoptosis more efficiently in these cells. Through the integration of nano-informatics and in vitro approaches, our study determined that PLA-PEG-PLA-CUR is an optimal system for delivering curcumin to inhibit cancer cells.

## 1. Introduction

The main challenge faced by the administration of hydrophobic anti-cancer drugs is their low solubility, resulting in poor biodistribution [1,2]. Encapsulation into nano-carriers can enhance the drugs’ selective delivery to their pharmacological site and decrease side effects [3]. Thus far, different types of materials [4] and nano-carriers have been studied for targeted delivery of developed drugs [5]. Polymeric nanomaterials offer certain advantages over other nano-carriers in terms of stability and modulation of drug release via controlling polymer biodegradation [6]. Polymeric nanomicelles containing both hydrophilic and hydrophobic moieties have the ability to self-assemble. Their properties can be precisely controlled and tailored for many applications. Also, their balance between the hydrophobic/hydrophilic ratio can be controlled, which is a prerequisite for the self-assembly process [7]. The amphiphilic nature of the block copolymers is an important characteristic for the drug delivery process [8].

According to previous observations, block copolymers enable the production of nanoscale materials [9,10] and provide the ability to engineer various nanostructures [11,12]. Block copolymers are macromolecules that contain two or more chemically distinct chains covalently bonded to each other for specific purposes [13]. The mixing entropy of block copolymers per unit volume is very small [14], which makes their thermodynamics discordant with other blocks [15]. These copolymers can be efficiently functionalized and used as potential drug delivery systems [16]. Furthermore, copolymers can be engineered to control drug loading and the release of drugs [17]. Block copolymers have two different types of drug-loading strategies [18]. In the first strategy, the drug is loaded into nanomicelles via covalent bonds between the drugs and the nanocarriers, which must be cleaved [19]. The second strategy is the most prevalent, in which the drug is entrapped inside micelles via physical interactions [20]. Here, therapeutic molecules are located within the core or shell of nanomicelles, and release often occurs via diffusion [21]. In fact, nanomicelles produced by the first and second strategies are structurally similar, but their stability attributes are different [22]. The stability of nanomicelles plays an important role in the safety and efficacy of these formulations [23].

This study was conducted to understand the biochemical behavior of PLA-PEG-PLA and PEG-PLA-PEG copolymers using in silico and in vitro approaches [24]. Herein, we employed molecular dynamics (MD) and free energy calculation to compare the dynamics and thermodynamics of understudied systems and compare them based on their ability to interact with the 1-palmitoyl-2-oleoyl-phosphatidylcholine (POPC) membrane. Also, curcumin-loaded PLA-PEG-PLA and PEG-PLA-PEG were investigated for their potential to deliver curcumin to MCF-7 cells. A variety of physicochemical properties, including zeta potential, size, and morphology, were assessed for PLA-PEG-PLA and PEG-PLA-PEG copolymers. In addition to measuring the loading capacity, we also assessed the release of curcumin from curcumin-loaded copolymers. The anticancer potential of curcumin-loaded copolymers was analyzed quantitatively using MTT, and the expression levels of Bcl-2, Bax, and hTERT genes were evaluated using RT-qPCR. Through a comprehensive comparison of nano-informatics and experimental results, this study achieved significant findings regarding the efficacy of designed biodegradable copolymers in loading CUR and inducing toxicity on a representative cancer cell line.

## 2. Materials and Methods

The work cycle utilized in this study included computational analysis of expected conformational behavior of copolymers, synthesis of copolymers, loading copolymers with curcumin, characterization of curcumin release in vitro, analysis of the effects of the curcumin-loaded polymers on viability of the MCF-7 cancer cells, and analysis of the effect of the treatment the MCF-7 cancer cells with the curcumin-loaded polymers on the expression several target genes. This work cycle is schematically represented in Figure 1.

### 2.1. Force Field

The morphology of the bilayer membrane and copolymer was generated using the CHARMM36 force field (an all-atoms forcefield). The CHARMM36 force field is a set of equations that describe the interactions between the atoms in these molecules. It is one of the most widely-used force field approaches for studying biological molecules and is known for its accuracy in predicting the structure and dynamics of biomolecules. This force field includes parameters for the bond lengths, bond angles, dihedral angles, and non-bonded interactions (such as van der Waals and electrostatic forces) between all pairs of atoms in a molecule. This approach provided a detailed understanding of the molecular interactions between the bilayer membrane and copolymer [25,26].

### 2.2. Copolymers and Biomembrane Preparation

The structures of PLA and PEG monomers were downloaded from *PubChem* in SDF format (Table 1). The downloaded monomers’ structures were optimized and linked together based on the planned design via HyperChem (v8.0.3) to form PLA-PEG-PLA and PEG-PLA-PEG copolymer molecules.

In the next step, the CHARMM-GUI server was utilized to generate a biomembrane with desired lipid compositions. This server is an online server owned by Lehigh University [27,28]. The CHARMM-GUI server possesses the capability to perform atomic-level characterization of both small molecules and large biomolecules. In the present study, we chose the pure 1-palmitoyl-2-oleoyl-phosphatidylcholine (POPC) bilayer. During biomembrane preparation, the rectangular box type was selected. The length of the Z-axis was adjusted in accordance with the hydration number, which specifies 33 water molecules per lipid molecule. The number of lipids was set to 64 on the upper and lower leaflet (overall 128 POPC molecules), and a surface area value of 63 Å was chosen. In the ionization step, 0.15 M NaCl was added to our system [29]. For membrane parametrization, force field CHARMM36 was selected. For the purpose of assessing their potential interaction with the target bilayer, the copolymers were positioned within a proximity of 5 Å from the membrane [24].

### 2.3. MD Simulations

MD is a useful approach for investigating the effect of solvent molecules on macromolecule structure and stability factors to obtain properties of the biomolecular systems, namely dipolar moment, density, conductivity, and diverse thermodynamic parameters, including entropies and interaction energies (Vandevals and GIBBS) [30]. In this study, we conducted the equilibration of the systems using the NVT (N = number of atoms, V = volume, and T = temperature) and NPT (P = pressure) ensembles. We utilized the CUDA-compiled GROMACS v2022.2 [31] software on Ubuntu v22.04 for this purpose. The temperature was maintained at approximately 310 K using the V-rescale model, with a time constant of 2 fs [32]. To control the pressure, we employed the Parrinello–Rahman algorithm with a compressibility value of 4.0 × 10^−5^ bar^−1^ [33]. Additionally, semi-isotropic coupling was applied to ensure a tensionless lipid bilayer. During the molecular dynamics (MD) simulation, the LJ potentials were adjusted to zero between a cutoff distance of 1.2 nm and rshift = 0.9 nm. The pair list was refreshed every 20 steps to ensure accurate and efficient calculations [34]. NVT and NPT steps were performed for 500 ps and 5 ns, respectively. In the production step, the configurations were sampled at every 100 ps. Finally, at the production step, all systems were simulated for 200 ns in TIP3P solvent [35]. After MD simulation, functions like root-mean-square deviation (RMSD), root-mean-square fluctuation (RMSF), solvent accessible surface area (SASA), radial distribution function (RDF), and mean square displacement (MSD) were extracted from the output trajectory files [33,36].

### 2.4. Binding Free Energy Analysis

The molecular mechanics–Poisson Boltzmann surface area (MM/PBSA) method was applied to compute the binding energy and impact of conformational changes on its value [37]. Here, the binding free energy was calculated using g_mmpbsa during the 200 ns of the molecular dynamics simulation [38]. During this step, the accessible surface area (SASA) model was utilized for apolar solvation energy calculation. The free energy computations of the POPC–copolymers binding are measured via the following Equation (1) [38]:(1)$${∆G}_{binding\;}=G_{complex\;}-(G_{POPC}+G_{copolymer})$$
where $G_{complex\;}$ is the total free energy of the POPC−copolymer, and $G_{POPC}$ and $G_{copolymer}$ denote total free energies of the POPC and copolymer in the solvent, respectively. Furthermore, the free energy for each individual entity can be given (2) [38]:(2)$$G_{binding}=(E_{MM})-\mathrm{TS}+(G_{solvation})$$

$G_{solvation}$ is the free energy of solvation, and T and S are the temperature and entropy, respectively. $E_{MM\;}$ is the average molecular mechanics potential energy in a vacuum, which is given as an Equation (3) [38]:(3)$$E_{MM}=(E_{bonded})+(E_{nonbonded})=(E_{bonded})+(E_{VdW\;}+G_{elec})$$
where $E_{elec}$ is electrostatic (to determine the stability and behavior of molecules) and $E_{vdw}$ is van der Waals interactions (weak attractive forces that arise between atoms and molecules of copolymers and POPC due to the fluctuations of their electron densities) between POPC and copolymers, and $E_{bonded\;}$ is bonded interactions consisting of angle, bond, and dihedral and unfit interactions. $\;E_{nonbonded\;}$ is the nonbonded interactions that include both van der Waals and electrostatic interactions. Furthermore, in the single-trajectory method, the conformation of membrane and copolymers in the bound and unbound forms are assumed to be the same. Therefore, $E_{bonded\;}$ is taken as zero, and $G_{solvation\;}$ is also calculated by Equation (4) [38]:(4)$$G_{solvation\;}=G_{polar\;}+G_{nonpolar}$$
where $G_{polar\;}and\;G_{nonpolar}\;$ are determined by polar solvation energy and SASA energy (a measure of the free energy associated with the exposure of a copolymer to a solvent), respectively.

### 2.5. Materials

Polyethylene glycol (PEG Mn = 4000), L-lactic acid (14.7 kDa), N, N_0_-dicyclohexylcarbodiimide, stannous 2-ethyl-hexanoate (SnOct_2_), 5-diphenyl-2-H-tetrazolium bromide (MTT), diethyl ether, dichloromethane, acetone, and ethanol were obtained from Sigma–Aldrich. The MCF-7 cell line was provided by the Pasteur Institute of Iran.

### 2.6. Synthesis of PLA-PEG-PLA Copolymer

The PLA-PEG-PLA block copolymer was synthesized from d, l-lactide with mPEG as the initial molecule and stannous octoate as the catalyst using a ring-opening polymerization (ROP) [39]. To begin polymerization, mPEG (1 mg), lactide (6 mg), and 0.05% (*w*/*w*) stannous octoate were heated to 130 °C in dichloromethane on continuous stirring. After 24 h, the produced polymer dissolved in chloroform and was cooled to room temperature. Then, it was precipitated in diethyl ether. In addition, for additional purification, the product was precipitated in distilled water three times. To dry the final products, the resulting polymer was first kept at room temperature for 24 h. Afterward, they were completely dried in a freeze dryer for 48 h −60 °C. The resulting product was a white powder, which was characterized and used for further studies. The final structure of the polymer was characterized by proton nuclear magnetic resonance spectroscopy (^1^H-NMR) in CDCL_3_ [40] at 500 MHz (Bruker Ac 500, Ettlingen, Germany) and Fourier transform infrared spectroscopy (FT-IR; Nicolet 550 A, Thermo Fisher Scientific, Waltham, MA, USA). Also, the surface charge and particle size of the PLA-PEG-PLA were determined by Dynamic Light Scattering (DLS; Malvern Zeta sizer 3000HS, Malvern, Worcestershire, UK). Additionally, the surface properties of block copolymers were determined using water contact angle (SDC100, Minder Hightech, Beijing, China) and a scanning electron microscope (SEM; JSM-5600LV, Jeol, Tokyo, Japan).

### 2.7. Synthesis of PEG-PLA-PEG Copolymer

As in the previous section, the synthesis of PEG-PLA-PEG copolymer was realized using ROP. mPEG (6 mg) was dissolved with lactide (1 mg) and 0.05% (*w*/*w*) stannous octoate in dichloromethane. The mixture was refluxed by continuous stirring at 110 °C for 18 h. Following solvent removal under vacuum, the resultant product was purified by dissolving in toluene and precipitating in cold ethyl ether (−20 °C) and then in distillate water for three times to obtain a purified product. Then, the copolymer was lyophilized for 48 h using conditions similar to the PLA-PEG-PLA copolymer. Similar to the previous section, the structure and surface characteristics of the synthesized copolymer were examined by FT-IR, NMR, DLS, and SEM.

### 2.8. Measurement of Curcumin-Encapsulation Efficiency

To measure curcumin-encapsulation efficiency in copolymers, 10 mg of freeze-dried PLA-PEG-PLA and PEG-PLA-PEG were dissolved in 10 mL acetone:water mixture (30:70 *v*/*v*) containing 1 mg of curcumin. The resulting solution was stirred for 30 min and then centrifuged at 8000 rpm for 10 min. Then, the supernatant was passed through a filter (14 kDa) and the curcumin-copolymer solution was obtained. Further, the freeze-drying process was implemented to prepare powders from the output of this step. Thereafter, the amount of loaded curcumin and encapsulation efficiency (EE) were calculated by measuring the absorbance of curcumin at 425 nm [41]. The following Equations (5) and (6) were used for calculating loading efficiency and encapsulation efficiency, respectively:(5)$$Drug\;loading(\%)=\frac{Weight\;of\;CUR}{Weight\;of\;CUR\;loaded\;Copolymer}\;\times \;100$$
(6)$$\mathrm{Encapsulation}\;\mathrm{efficiency}\;(\%)=\frac{Weight\;of\;entrapped\;CUR\;in\;Copolymer}{Weight\;of\;feeding\;CUR\;}\times \;100$$

The drug loading and encapsulation were also calculated in different weight ratios of drug and copolymer (CUR:copolymer; 1:10, 1:20, and 1:50).

### 2.9. In Vitro Release Study

The dialysis bag diffusion technique was applied to study the curcumin release behavior from the copolymers. The freeze-dried curcumin-copolymers with curcumin: copolymer ratios of 1:10, 1:20, and 1:50 were analyzed for release studies. Briefly, the samples containing 20 mg of curcumin-loaded block copolymers were poured into a dialysis bag (Mw 14 kDa). The dialysis bag was then suspended in 40 mL of phosphate-buffered saline (PBS, 0.01 M, pH 7.4) and kept in an incubator shaking mixer (Taitec, BR-42FL, Koshigaya City, Japan) at 100 rpm and 37 °C. Samples were removed from the receptor medium at time intervals of 1, 2, 3, 4, 6, 8, 10, and 100 h. They were then replaced with the same volumes of fresh PBS. The released curcumin value was quantified using its absorbance at 425 nm using a UV–vis spectrophotometer.

### 2.10. Cell Culture

MCF-7 cells were cultured at 37 °C in a humidified atmosphere (95% air, 5% CO_2_) in RPMI-1640 medium with 10% (*v*/*v*) FBS, 100 gmL^−1^ streptomycin, 100 UmL^−1^ penicillin, 2 mM L^−1^ glutamine, and 1% non-essential amino acid.

### 2.11. In Vitro Cell Viability Assay

MCF-7 breast cancer cells were employed to evaluate the cytotoxicity of block copolymers and those containing curcumin using MTT essay. MCF-7 cells were seeded in 96-well plates (5 × 10^3^ cells per well) with 150 µL/well RPMI medium containing 10% FBS and 1% penicillin/streptomycin. Cells were incubated for one day at 37 °C in a humidified incubator with 5% CO_2_ and 95% humidity. In all cellular assays, the cells were treated with different concentrations (0–60 µM) of free curcumin, PLA-PEG-PLA, PEG-PLA-PEG, curcumin-PLA-PEG-PLA, and PEG-PLA-PEG-curcumin, for 48 h. PBS-treated cells were taken as control (cnt). Percent cell death was calculated by following Equation (7), and the IC50 values for curcumin, curcumin-PLA-PEG-PLA, and curcumin-PEG-PLA-PEG were determined [42].
(7)$$\mathrm{Cell}\;\mathrm{death}\;(\%)=\frac{Absorption\;value\;of\;treated\;cells-Absorption\;value\;of\;backghround}{Absorption\;value\;of\;non-treated\;cells-Absorption\;value\;of\;background}\times 100$$

### 2.12. Gene Expression Studies

MCF-7 cells were cultured in six-well plates for 24 h. Thereafter, the cells were treated with free curcumin and copolymers–formulated curcumin at their IC_50_ concentrations. Control cells were treated with PBS. For gene expression analysis, the cells were harvested after 48 h of treatment. Total RNA was extracted at each time point utilizing the PARS RNA extraction kit (Pars-Tous, Mashhad, Iran). RNA concentration and purity were quantified using a UV–vis spectrophotometer (NanoDrop^TM^ One^C^, Thermo Fisher Scientific, Waltham, MA, USA). cDNA synthesis was performed with the PARS cDNA synthesis kit (Pars-Tous, Mashhad, Iran) using the manufacturer’s protocol. To examine the expression level of Bax, Bcl_2_, and hTERT, the primers listed in Table 2 were used. Each PCR reaction contained 2 µL of cDNA, 2 µL of primers, 6 µL DEPC treated water, and 10 µL Amplicon SYBR Green master mix (Ampliqon A/S, Odense, Denmark). The Rotor-Gene 6000 Real-Time PCR device (Corbett Research, Hilden, Germany) was utilized for gene expression studies. The cycle threshold (Ct) values for each gene were normalized using the Ct value of GAPDH. The relative expression of the genes was quantified by using the 2^−ΔΔCt^ method.

### 2.13. Statistical Analysis

The statistical difference between the quantitative data such as contact angles, MTT assay, and gene expression was determined using analyses of variance (one-way ANOVA). *p* < 0.05 was considered significant when comparing the means. Statistical Package for the Social Sciences (SPSS) software, version 21 (SPSS Inc., Chicago, IL, USA), was used for statistical analyses.

## 3. Results

### 3.1. Preliminary Molecular Dynamics Analysis

Figure 2 shows the RMSD values over time for the block copolymers under investigation. The graph reveals how the RMSD initially increases rapidly and then levels off, indicating that the system has equilibrated and reached a stable conformational state. The magnitude of the RMSD also provides a measure of the extent of conformational changes occurring in the system. In this way, the RMSD analysis provides valuable insights into the dynamics and equilibration of block copolymer systems, which are crucial for designing and optimizing materials with specific properties. As a result, there is a smaller RMSD for PEG-PLA-PEG than PLA-PEG-PLA. However, there was no significant difference in RMSD values between the two block copolymers, and the conformations of copolymers did not change noticeably. Furthermore, RMSD averages for PLA-PEG-PLA and PEG-PLA-PEG were 0.17634 and 0.168796 nm, respectively.

A further significant measurement is the RMSF value, which we calculated during 200 ns of molecular dynamics simulation. The RMSF analysis revealed important information about the dynamics and stability of the block copolymer systems. For example, regions with high RMSF values corresponded to flexible or disordered regions, while regions with low RMSF values corresponded to more rigid or ordered regions. This information was used to identify important structural features of the block copolymer and to understand how they affect the material properties.

The comparative investigation of RMSF values displays that PLA-PEG-PLA fluctuated between 0.14–0.18 nm, while the PEG-PLA-PEG value was between 0.05–0.15 nm (Figure 3).

### 3.2. Prediction of Block Copolymer Properties

For the analysis of water penetration into the POPC membrane, RDF was enforced. Overall, the use of RDF analysis in our study allowed us to gain a deeper understanding of the behavior of the POPC membrane in the presence of water molecules. The highest peak corresponds to the maximum value of the density variation from the reference of water molecules at that distance. As seen in Figure 4, this peak for copolymers is close and very sharp at 0.515 nm, and this peak shows the distance where the most water molecules are available. A smaller distance and a larger g (r) indicated more penetration of water into the membrane.

SASA can measure the proportion of the copolymers’ surface that can be accessed by the water molecules. In other words, in this case, SASA can be used to measure the proportion of the copolymer surface that can be accessed by water molecules. These data can confirm RDF results conducted by the MD trajectory. The average SASA amount for PLA-PEG-PLA was 8.0923 nm^2^, while the average SASA amount for PEG-PLA-PEG was 3.7654 nm^2^. From Figure 5, we can see that PLA-PEG-PLA had higher SASA values over a span of 200 ns.

The contact number of copolymers with POPC can reflect the surface physicochemical properties of copolymers on their probable localization. Figure 6 shows that PLA-PEG-PLA has more contacts with the POPC. The average number of contacts between PLA-PEG-PLA and PEG-PLA-PEG with membrane are 1000 and 500, respectively.

The use of contact number analysis in this study showed the surface properties of block copolymers and their interactions with lipid membranes. By combining this analysis with other molecular dynamics simulations and experimental measurements, we can develop a more complete picture of the properties and behavior of block copolymer systems in complex environments.

The distance analysis between the block copolymers and the biomembrane surface can be considered as an appropriate parameter to identify the interaction between the copolymers and the POPC surface. Figure 7 exhibits the minimum distance of the center of mass (COM) of the block copolymers and membrane surface. The averages of distances were 0.85 nm for the POPC/PLA-PEG-PLA system and 0.94 nm for the POPC/PEG-PLA-PEG.

We achieved a deeper understanding of the factors that influence copolymer–membrane interactions and that can help identify potential binding sites or regions of interest for further investigation by analyzing the distance between the copolymer and different regions of the membrane surface. In particular, a larger g (r) value at smaller distances suggests a higher probability of finding water molecules in the immediate vicinity of lipid headgroups, indicating a stronger interaction between water molecules and lipid headgroups. This may result in disrupted lipid packing and an increase in lipid mobility or permeability.

### 3.3. Membrane Penetration

An MSD function was calculated to evaluate the penetration ability of the studied copolymers in the POPC membrane. This involved tracking the position of the copolymers over time and calculating the average squared displacement from their initial position. The resulting MSD function provided information about the diffusion behavior of the copolymers and their ability to penetrate the membrane. Moreover, in this case, two strategies were considered to demonstrate copolymer diffusion: transverse and lateral diffusion. The average of transverse diffusion (Figure 8A) for the PLA-PEG-PLA (3.67895 nm^2^) and PEG-PLA-PEG (2.4654 nm^2^) showed that the membrane penetration of PLA-PEG-PLA is much higher. However, the average of lateral diffusion for the PLA-PEG-PLA (14.15 nm^2^) and PEG-PLA-PEG (23.67 nm^2^) showed that the membrane penetration of PEG-PLA-PEG is much higher. As shown in Figure 8B, the PLA-PEG-PLA and PEG-PLA-PEG graph explains that at about 100 ns, both copolymers had almost the same speed of penetration into the membrane and had the same lateral penetration. After 100 ns, the function of the copolymers for lateral diffusion was different, and the lateral diffusion of the PEG-PLA-PEG molecule increased with a very steep slope in the polar area of the membrane, and PLA-PEG-PLA continued to propagate laterally with a much lower slope. At 180 ns, both copolymers were entrapped in the membrane and had fluctuation. Eventually, they resumed their lateral diffusion after a few nanoseconds. From the obtained results, the higher mobility of the PEG-PLA-PEG molecule is clear in lateral diffusion. This finding is justified by following the molecule’s mobility around the head of the POPC membrane, and it should be noted that the non-polar area of the PEG-PLA-PEG molecule is smaller, and the polar area is more uniform than the PLA-PEG-PLA molecule (Figure 8C).

### 3.4. Energy Calculations

Binding free energy is one of the most important and effective parameters for analyzing how the components interacted with each other inside the simulated box. Furthermore, binding free energy is proven to be a powerful tool for analyzing the interactions between different components in molecular simulations. Thus, free energy was measured using the MM/PBSA method for extracted snapshots at every 250 ps of intervals from MD trajectories. Table 3 represents the results of MM/PBSA calculations.

The binding free energy from Equation (2) is −3003.701 and −2797.846 $\frac{kj}{kmol}$ for PLA-PEG-PLA and PEG-PLA-PEG, respectively. Moreover, the average amount of potential energy for the system containing PLA-PEG-PLA and PEG-PLA-PEG is −287,450 and −256,855 $\frac{kj}{kmol}$, respectively, which indicates that the system containing PLA-PEG-PLA has greater stability.

### 3.5. Characterization of Synthesized Copolymers

The chemical structure and composition of the synthesized block copolymers were analyzed using ^1^H-NMR and FT-IR. In the FT-IR analysis, peaks at 1698.03 and 1728.34 cm^−1^ indicated the presence of the carboxylic acid group in lactic acid, which was more prominent in PLA-PEG-PLA. Additionally, peaks at 2875.44 and 2870.54 cm^−1^ corresponded to (C-H) in the formation of the copolymer. The ^1^H-NMR spectra of the samples dissolved in CDCL_3_ confirmed the formation of block copolymers, with peaks at 1.56 ppm and 5.17 ppm attributed to the methyl (CH_3_) and methine (CH) groups of lactic acid, respectively, in the PLA-PEG-PLA copolymer. Peaks at 3.64 ppm and 3.48 ppm referred to protons of the methylene group (CH_2_) and methoxy end groups (CH_3_O) in the PEG blocks, respectively, and peaks at 4.3 ppm corresponded to methylene protons attached to methine groups in lactide monomers at the end of PEG chains. In the ^1^H-NMR spectra of PEG-PLA-PEG, peaks at 5.17 ppm and 3.64 ppm were assigned to (CH) of PLA and (CH_2_) of PEG, respectively (Figure 9). These analyses provide valuable insights into the chemical structure and composition of the synthesized copolymers, which can inform their potential applications as drug delivery systems.

Particle size, surface morphology, and zeta potential are shown in Figure 10. The copolymer sizes were determined by DLS. The copolymers had a uniform particle size distribution. The diameter and zeta potential of PEG-PLA-PEG were 249.4 nm and −7.8 ± 0.1 mV, respectively. Also, the particle size for PLA-PEG-PLA increased to 267.3 nm compared to PEG-PLA-PEG. The zeta potential of this sample was −1.3 ± 0.09 mV. However, the size of the PEG-PLA-PEG observed by DLS was smaller than PLA-PEG-PLA, and the charge of this copolymer is more negative. Additionally, the conjugation of polymers also influences their shape.

As shown in (Figure 10D,E), the morphology of both the block copolymers has a regular spherical structure and a homogeneous surface without signs of collapse. However, the surface of PEG-PLA-PEG is slightly more swollen than that of PLA-PEG-PLA, which can appear during the drying step, and at the same magnification, it has a smaller diameter.

Another important feature of the copolymer surface is the degree of hydrophilicity that is displayed by contact angle measurement. The results of this analysis are shown in Figure 11 and were 19.234 and 13.456° for PLA-PEG-PLA and PEG-PLA-PEG, respectively. Since their chain terminals are different, they have distinct degrees of hydrophilicity. When curcumin was added to the systems, the extent of hydrophilicity decreased and was reported as 62.751 for curcumin-PLA-PEG-PLA and 47.312 for curcumin-PEG-PLA-PEG.

### 3.6. Encapsulation Efficiency and Drug Loading

The drug loading (DL) and encapsulation efficiency (EE) of copolymers and curcumin are strongly influenced by the *w/w* ratio. In this study, the encapsulation efficiency for PLA-PEG-PLA was 43.9 ± 2.1% at a *w*/*w* ratio of 1:50. However, at a ratio of 1:10, the drug encapsulation efficiency increased to 61.4 ± 1.4%. Similarly, for PEG-PLA-PEG, the encapsulation efficiency at a *w*/*w* ratio of 1:50 was 36.2 ± 1.9%, which increased to 52.3 ± 1.3% at a ratio of 1:10 (Table 4). These findings emphasize the importance of optimizing the *w/w* ratio in copolymer-based drug delivery systems to achieve maximum drug loading and encapsulation efficiency, which can ultimately improve their therapeutic efficacy (Table 4).

### 3.7. In Vitro Drug-Release Study

The release profiles of curcumin from curcumin-loaded copolymers were evaluated to assess their potential as carriers for the drug. The curcumin release profiles for curcumin-loaded PLA-PEG-PLA and PEG-PLA-PEG with different *w*/*w* ratios are displayed in Figure 12. A sustained release of curcumin was observed for 100 h by diffusing through the PLA-PEG-PLA copolymer compared with the free curcumin, which released 92% within 40 h. Initially, about 20% of the loaded drug rapidly was released in 18 h for two systems (curcumin-loaded PLA-PEG-PLA and curcumin-loaded PEG-PLA-PEG); then, the drug was released slowly for the remaining time. Approximately 80% of the total curcumin loaded in PLA-PEG-PLA at 1:10 *w*/*w* ratio was gradually released over 100 h. The PLA-PEG-PLA released 60.3% and 53.8% of the total curcumin, respectively, at *w*/*w* ratios of 1:20 and 1:50. The release profile of PEG-PLA-PEG shows the same mechanism as the PLA-PEG-PLA. After 18 h, about 20% of the loaded drug was released, and then, curcumin was released slowly up to 100 h. Approximately 88% of the curcumin loaded in PEG-PLA-PEG at a 1:10 *w*/*w* ratio was released over 100 h, which had a faster release than PLA-PEG-PLA. Also, at *w*/*w* ratios of 1:20 and 1:50, the total curcumin released was 68.2% and 59%, respectively (Figure 12).

### 3.8. Cell Viability Studies

The effect of curcumin-loaded block copolymers on MCF-7 cell viability was studied using an MTT assay. Cell death increased with increasing concentration of curcumin (Figure 13). The results indicated that PLA-PEG-PLA-curcumin reduced the cancer cell viability to a greater extent compared to PEG-PLA-PEG-curcumin and free curcumin. The IC_50_ value of curcumin was 29.8 µM ± 0.61, while this concentration (29.8 µM) of curcumin delivered through curcumin-PLA-PEG-PLA and curcumin-PEG-PLA-PEG increased the cell death to 62.3% ± 0.79 and 57.6% ± 0.53, respectively. The IC_50_ values for curcumin-PLA-PEG-PLA and curcumin-PEG-PLA-PEG were calculated to be 23.01 ± 0.85 and 26.87 ± 0.49 µM, respectively. The cell death at 60 µM curcumin equivalent after 48 h was reported to be 80.4 ± 0.99 for curcumin, 85.3 ± 0.76 for curcumin-PEG-PLA-PEG, and 91.1 ± 1.01 for curcumin-PLA-PEG-PLA. These results suggested that curcumin-PLA-PEG-PLA is the most effective system for inducing cancer cell death.

### 3.9. Gene Expression Analysis

The expression of the Bax gene increased and was found to be 2.4-fold higher for curcumin-PLA-PEG-PLA, 2.1-fold higher for curcumin-PEG-PLA-PEG, and 1.3-fold higher for curcumin-treated cells compared to control cells (Figure 14A). As shown in Figure 14B, gene expression analysis showed a significant reduction in Bcl-2 levels in curcumin-copolymers and curcumin-treated cells after 48 h of treatment. The Bcl-2 fold change for curcumin, curcumin-PLA-PEG-PLA, and curcumin-PEG-PLA-PEG was 0.85, 0.38, and 0.49, respectively. Also, the expression level of the hTERT gene was reduced in curcumin-, curcumin-PLA-PEG-PLA-, and curcumin-PEG-PLA-PEG-treated cells, and the fold change in expression was 0.78-, 0.34-, and 0.41-fold compared to PBS-treated cells, respectively (Figure 14).

## 4. Discussion

The study of nanocarrier behavior is essential to ensure steady and controlled drug release at the target site. Hence, the determination of structural properties, surface characterization, and permeation tendency into the biomembrane is critical for designing carriers such as copolymers for the delivery of drugs [42,43,44]. The extent of hydrophilicity and hydrophobicity is an important characteristic of the copolymers, which can be approximately accessed via contact angle [45,46]. For the analysis of water penetration into the POPC membrane by MD simulation, RDF was enforced so that the highest peak corresponds to the maximum value of the density variation from the reference of water molecules at that distance [47]. The peak for copolymers was close and very sharp at 0.385 nm, and the penetration of water molecules into the POPC was high. However, there was a quantitative difference in the average RDF for both copolymers, so the amount of accumulation of water molecules around the membrane was higher in the presence of PEG-PLA-PEG.

These results may point to the greater polarity of PEG-PLA-PEG compared to PLA-PEG-PLA copolymer, as shown by the homogeneous red areas of the PEG-PLA-PEG in Figure 8C. Therefore, we expected a more negative value of zeta potential for PEG-PLA-PEG, which was then confirmed in Figure 10A, indicating that the surface charge of the PEG-PLA-PEG was more negative compared to PLA-PEG-PLA. However, the RDF and zeta results gave limited information about the hydrophilic properties of copolymers. Thus, the contact angle was compared for each system. It was unexpected that the simulation predictions followed the experimental data. PEG-PLA-PEG had a smaller contact angle, and it can be argued this copolymer is more hydrophilic than PLA-PEG-PLA.

MSD analysis was used to obtain more accurate results from the copolymer’s penetration into the membrane. In this case, two theories have been considered to demonstrate copolymer penetration: transverse and lateral diffusion. Each theory shows a different function of copolymers. Figure 8 illustrates the differences between the two. Lateral diffusion is greater for the PEG-PLA-PEG than the PLA-PEG-PLA molecule, suggesting that the PEG-PLA-PEG molecule is more active across the hydrophilic area of the membrane, and PLA-PEG-PLA has better penetration along the Z-axis.

At transverse diffusion, the highest peaks occur at times 48, 101, 150, and 190 ns, which is longer for PLA-PED-PLA than PEG-PLA-PEG. The greater transverse diffusion of molecule PLA-PEG-PLA and its ups and downs along the Z-axis can depend on two factors that are explained in Figure 8C. This behavior can be described based on the length of the carbon chain in the copolymers. The PLA-PEG-PLA molecule (C_8_H_14_O_6_) has a longer chain of carbon, while the PEG-PLA-PEG molecule (C_7_H_14_O_5_) has a shorter chain. Hence, the non-polar chain causes more interaction with the hydrophobic area of the POPC membrane and penetrates more easily. The second reason for the greater PLA-PEG-PLA transverse penetration is related to the polar region of the molecules. As shown in Figure 8, the polar regions of the PEG-PLA-PEG molecule exhibit a more homogeneous distribution. This can be considered the key to the dynamism of the PEG-PLA-PEG molecule in the polar area of the membrane laterally and in trying to move and stay in the width of the membrane. Overall, differences in transverse and lateral diffusion between different block copolymers affect their interactions with the lipid membrane and their potential applications. Block copolymers that exhibit higher transverse diffusion may have a greater ability to penetrate the membrane and deliver drugs to the interior of cells, while block copolymers with higher lateral diffusion display enhanced membrane remodeling or fusion capabilities. Additionally, block copolymers with higher transverse diffusion are more susceptible to membrane disruption or destabilization in the presence of certain solvents or stresses, while block copolymers with higher lateral diffusion are more resistant to such effects.

Further, contact number and distance illustrate the interaction between the tail of the copolymers chain and the membrane surface. The oxygen atom in the PLA-PEG-PLA chain tail has more electronegativity than the carbon atom in the PEG-PLA-PEG chain tail. A high affinity to interact with the hydrogen atoms on the POPC and more contact with the membrane surface can be correlated with better penetration of PLA-PEG-PLA. Also, the long distance indicates weak interaction and lower binding affinity between the copolymers and the membrane. The systems containing PLA-PEG-PLA have a small-distance membrane throughout the MD simulation. As a result of cell penetration and the tendency of PLA-PEG-PLA to connect more strongly to the cell membrane, this copolymer should be more effective as a delivery agent, and also, the release of curcumin is slower from this copolymer owing to its higher hydrophobic content, which is also demonstrated by curcumin-release studies. Due to the higher hydrophobicity of PLA-PEG-PLA over PEG-PLA-PEG, this copolymer offers more chances for drug release from its hydrophobic core. Moreover, the drug-release profile explained that the burst release of the PLA-PEG-PLA copolymer was less than the other two systems, and a slower drug release was observed in this copolymer. Different hydrolysis-induced degradation rates of polymers may explain such differences in drug-release profiles. Hence, a higher ratio of hydrophilic PEG to hydrophobic PLA caused polymer destruction and higher drug release at initial time points in PEG-PLA-PEG systems, and the curcumin-release profile from PLA-PEG-PLA was more controlled.

Moreover, free energy binding results express some facts about nanocarriers. Free energy binding values, which indicate better PLA-PEG-PLA performance, show a tendency for this copolymer to be more dynamic on the Z-axis of the membrane. Hence, the results of type diffusion and lateral diffusion can be explained. Considering that curcumin is a hydrophobic drug and in order to facilitate its release [48], a hydrophilic carrier with a hydrophobic core such as PLA-PEG-PLA can improve the effectiveness of this compound.

Favorable attributes of PLA-PEG-PLA, including higher drug loading and slow release of loaded curcumin, are likely the key contributors to increased cell death in MCF-7 cells. Further, this copolymer affected the gene expression levels to the greatest extent compared to curcumin and curcumin-PEG-PLA-PEG. The decrease in Bcl-2 and hTERT expression and increase in Bax expression correlates with apoptosis-mediated death of MCF-7 cells [49,50,51]. Overall, our results indicated that PLA-PEG-PLA is an efficient delivery vehicle for hydrophobic drugs such as curcumin.

## 5. Conclusions

The primary objective of this study was to investigate the ability of linear triblock copolymers (PLA-PEG-PLA and PEG-PLA-PEG) to interact with cell membranes and deliver curcumin using a combination of MD simulations and in vitro experiments. The MD simulations revealed that the PLA-PEG-PLA copolymer exhibited higher interaction potency with the POPC membrane compared to PEG-PLA-PEG. Additionally, our energy analysis demonstrated that PLA-PEG-PLA is more stable than PEG-PLA-PEG, indicating that it should be a promising candidate for the intracellular delivery of hydrophobic drugs. Experimental analysis including ^1^H-NMR and FT-IR confirmed the successful synthesis of the copolymers. The copolymer size, charge, and morphology were consistent with the simulation results. Furthermore, PLA-PEG-PLA exhibited higher drug loading and more controlled release rates, leading to greater cell death in MCF-7 cells and a more significant effect on the expression of target genes. Overall, this study provides important insights into the behavior and properties of linear triblock copolymers and their potential applications in drug delivery. These findings suggest that PLA-PEG-PLA should be an effective candidate for the delivery of CUR and other hydrophobic drugs.

## Figures and Tables

**Figure 1 polymers-15-03133-f001:**
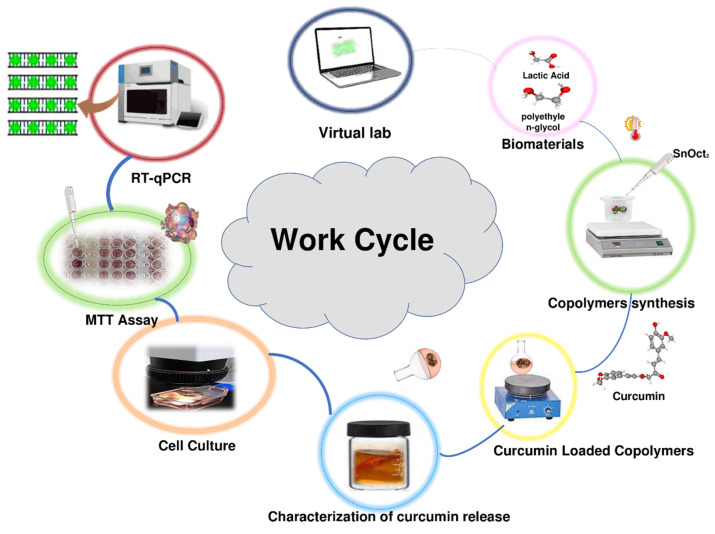
Schematic representation of the work cycle utilized in this study.

**Figure 2 polymers-15-03133-f002:**
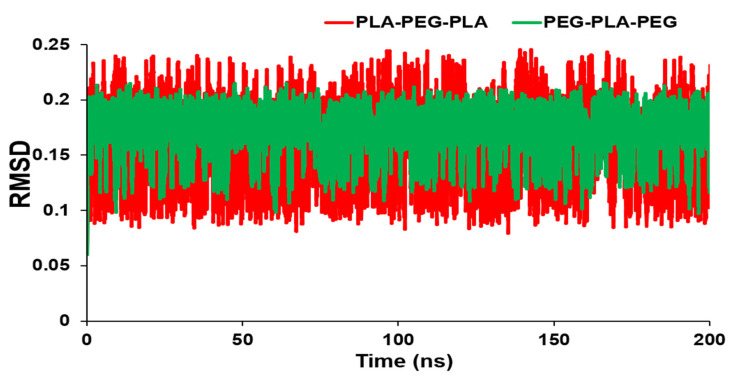
RMSD values of PLA-PEG-PLA and PEG-PLA-PEG.

**Figure 3 polymers-15-03133-f003:**
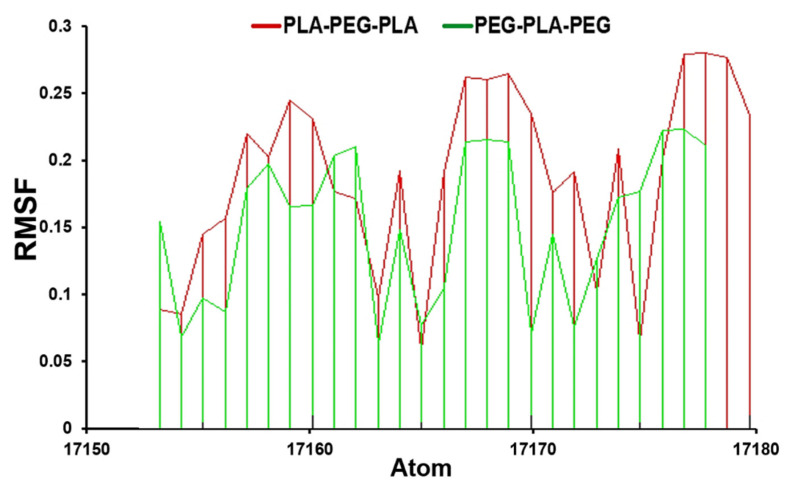
RMSF analysis for PLA-PEG-PLA and PEG-PLA-PEG.

**Figure 4 polymers-15-03133-f004:**
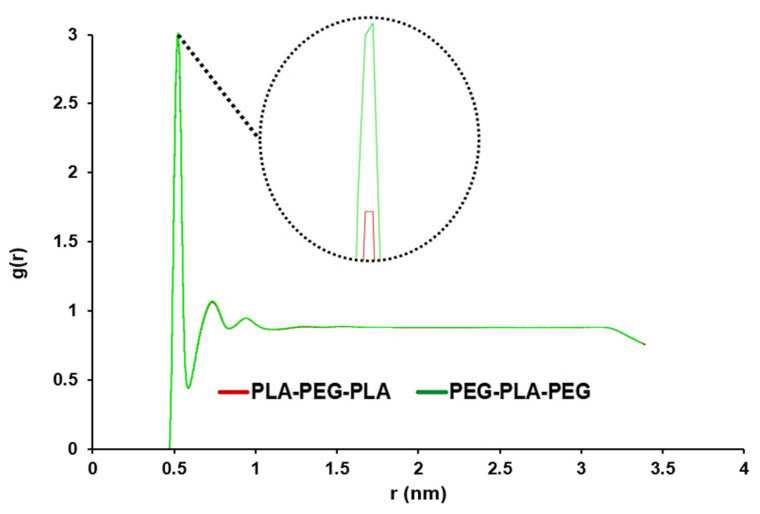
Comparative RDF to analyze the water penetration capability to pass through POPC in PLA-PEG-PLA and PEG-PLA-PEG systems (the graphs of the designed systems aligned with each other).

**Figure 5 polymers-15-03133-f005:**
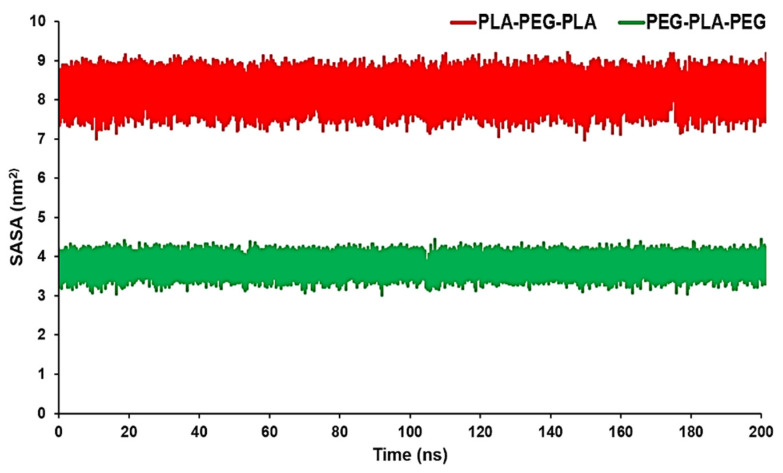
SASA analysis for PLA-PEG-PLA and PEG-PLA-PEG.

**Figure 6 polymers-15-03133-f006:**
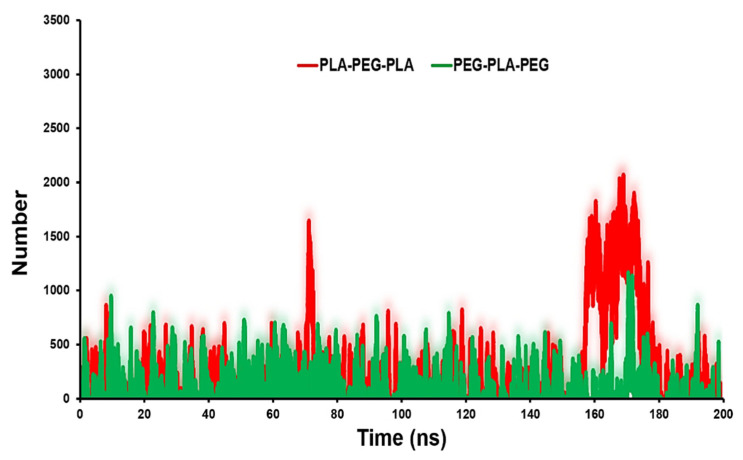
Contact number analysis for PLA-PEG-PLA and PEG-PLA-PEG to POPC membrane.

**Figure 7 polymers-15-03133-f007:**
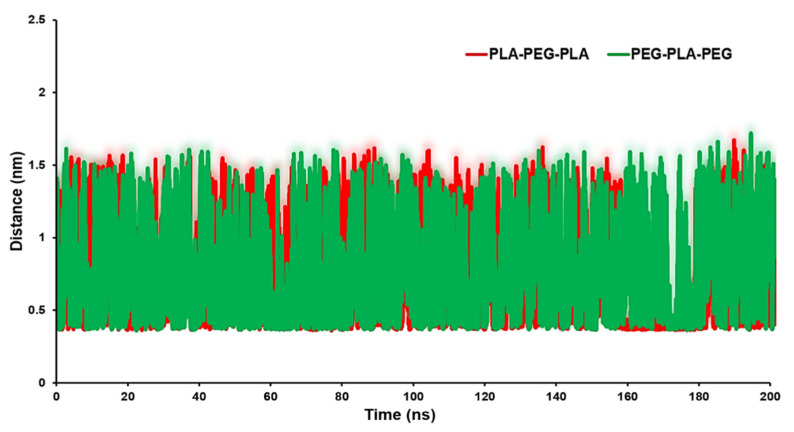
Distance between target copolymers and POPC membrane.

**Figure 8 polymers-15-03133-f008:**
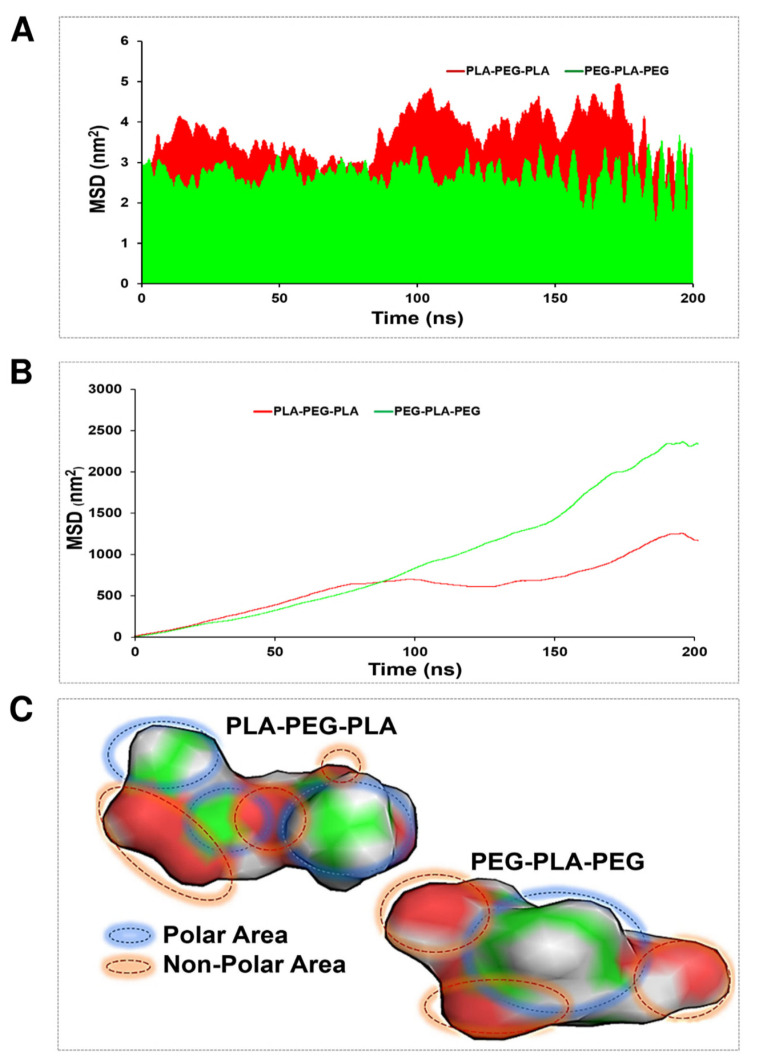
MSD evaluation for PLA-PEG-PLA and PEG-PLA-PEG. (**A**) Transverse diffusion, (**B**) lateral diffusion, and (**C**) polar and non-polar areas of block copolymers via Gauss View.

**Figure 9 polymers-15-03133-f009:**
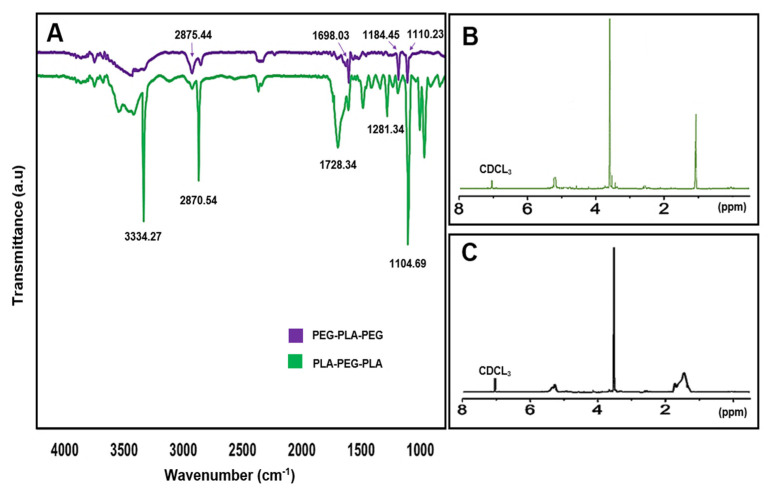
(**A**) The FT-IR spectra of copolymers, (**B**) ^1^H-NMR spectra of PLA-PEG-PLA, and (**C**) ^1^H-NMR spectra of PEG-PLA-PEG.

**Figure 10 polymers-15-03133-f010:**
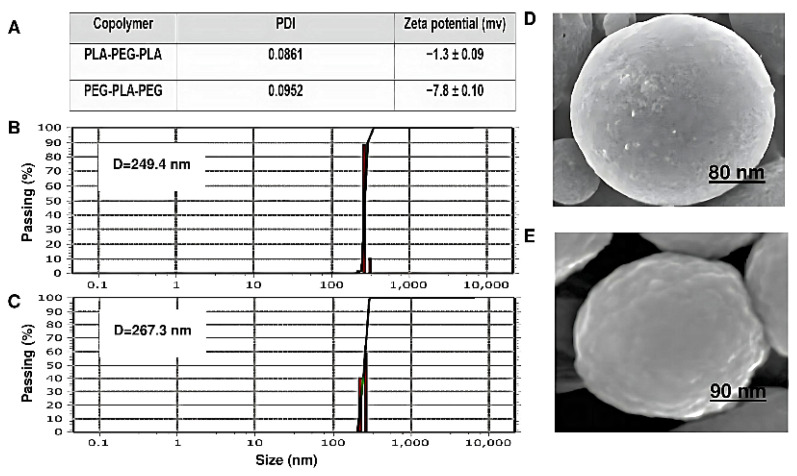
Characterization and morphology investigation. (**A**) Zeta potential and polydispersity index (PDI) measurements of copolymers, (**B**) average particle size diameter of PEG-PLA-PEG, (**C**) average particle size diameter of PLA-PEG-PLA, (**D**) SEM images of PLA-PEG-PLA, and (**E**) SEM images of PEG-PLA-PEG.

**Figure 11 polymers-15-03133-f011:**
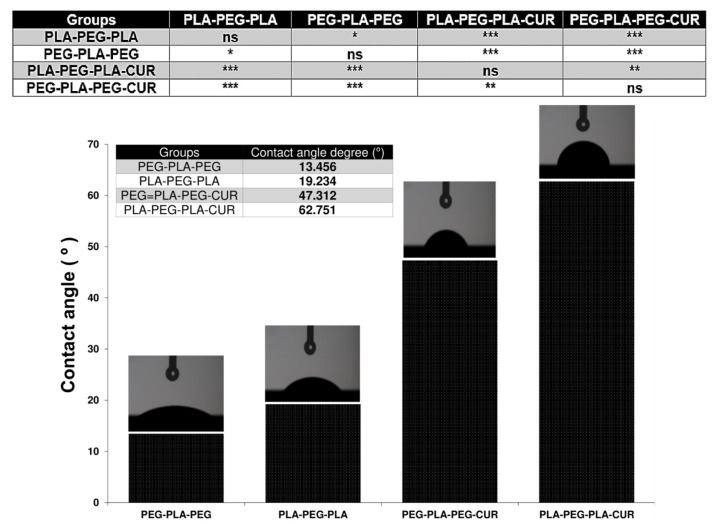
Contact angle measurements. Values represent the mean ± SD, and data were analyzed using one-way ANOVA (*n* = 4). * *p* < 0.05, ** *p* < 0.01, and *** *p* < 0.001.

**Figure 12 polymers-15-03133-f012:**
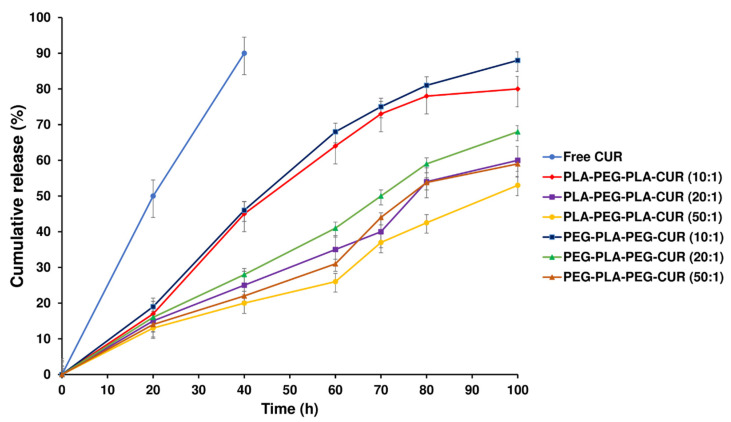
In vitro release of free curcumin and curcumin-loaded PLA-PEG-PLA and PEG-PLA-PEG at various *w*/*w* ratios.

**Figure 13 polymers-15-03133-f013:**
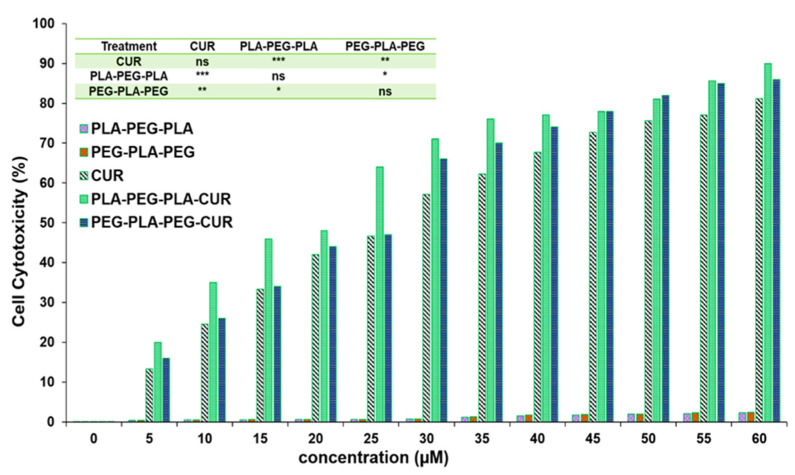
Evaluating the inhibitory effect of curcumin-copolymers on MCF-7 cancer cells at 48 h. Values represent the mean ± SD, and data were evaluated using one-way ANOVA (*n* = 3). * *p* < 0.05, ** *p* < 0.01, and *** *p* < 0.001.

**Figure 14 polymers-15-03133-f014:**
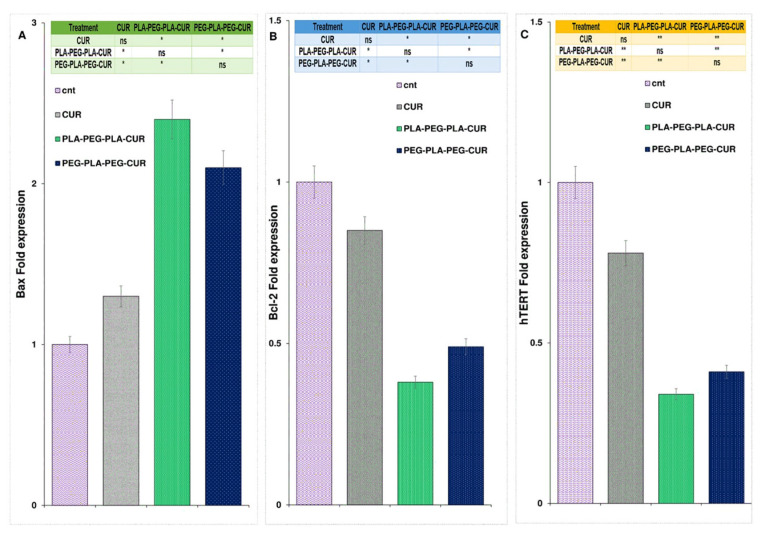
The figure illustrates the alterations in the expression of Bax (**A**), Bcl-2 (**B**), and hTERT (**C**) genes following the administration of curcumin and the designed copolymers over a 48-hour period. Gene expression analysis was conducted using one-way ANOVA with a sample size of three (*n* = 3). Statistical significance levels are denoted as * *p* < 0.05 and ** *p* < 0.01.

**Table 1 polymers-15-03133-t001:** Target molecules along with their chemical features.

Name	Molecular Formula	3D Structure	PubChem ID
DL-Lactic acid	CH_3_CHOHCOOH	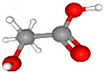	612
Ethane-1,2-diol	HOCH_2_CH_2_OH	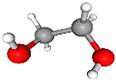	174

**Table 2 polymers-15-03133-t002:** Designed primers information.

Gene Name	Primer Sequence	Strand	Product Size
BAX	AAACTGGTGCTCAAGGCCC	Plus	81
	CCGGAGGAAGTCCAATGTCC	Minus	
BCL2	TGGGATCGTTGCCTTATGCA	Plus	101
	GTCTACTTCCTCTGTGATGTTGT	Minus	
hTERT	AACCTTCCTCAGGACCCTGG	Plus	128
	CCGGCATCTGAACAAAAGCC	Minus	
GAPDH	TGGAAGGACTCATGACCACA	Plus	119
	AGAGGCAGGGATGATGTTCT	Minus	

**Table 3 polymers-15-03133-t003:** Binding free energy terms information for studied systems.

Copolymer	Energy (kJ/mol)				
	Van der Waal Energy	Electrostatic Energy	Polar Solvation Energy	SASA Energy	Entropy
PLA-PEG-PLA	−11.203	−5.649	36.657	−1.936	9.747
PEG-PLA-PEG	−3.859	−2.884	27.055	−0.878	9.088

**Table 4 polymers-15-03133-t004:** Drug loading (DL) and encapsulation efficiency (EE) of curcumin at various drug–polymer ratios.

Systems	Ratio (*w*/*w*) CUR: Copolymers	DL (%)	EE (%)
Curcumin-PLA-PEG-PLA	1:50	1.9 ± 0.3	43.9 ± 2.1
	1:20	4.4 ± 0.1	51.7 ± 2.5
	1:10	6.3 ± 0.2	61.4 ± 1.4
Curcumin-PEG-PLA-PEG	1:50	1.2 ± 0.1	36.2 ± 1.9
	1:20	3.6 ± 0.3	40.3 ± 2.8
	1:10	4.8 ± 0.6	52.3 ± 1.3

## Data Availability

The authors confirm that the data supporting the findings of this study are available within the article.

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
