# Peer review of "Design, Synthesis, and Comparison of PLA-PEG-PLA and PEG-PLA-PEG Copolymers for Curcumin Delivery to Cancer Cells"

_polymers, 2023, doi:10.3390/polym15143133_

Round 1

Reviewer 1 Report

Reviewers' comments:

Manuscript Number: polymers-2487179

Title: Design, synthesis, and comparison of PLA-PEG-PLA and PEG-PLA-PEG copolymers for curcumin delivery to cancer cells

Comments:

This work concerns the Design, synthesis, and comparison of PLA-PEG-PLA and PEG-PLA-PEG copolymers for curcumin delivery to cancer cells. Important points are missing and there are some points that should be revised or corrected. Some important points are mentioned hereafter.

- In the Abstract, the authors need to improve with more specific short results and conclusions, i.e. academic novelty or technical advantages.

- Keywords: add more keywords.

- In the introduction, the authors do not show the significance and novelty of the work.

- Give more detail for the Evaluation of 2.1. Force field.

- CH3CHOHCOOH……..to…………. CH3CHOHCOOH

- HOCH2CH2OH……to………….. HOCH2CH2OH

- Please provides the references for equations and formula.

- In part SEM: how the energy of the accelerator beam used?

- 3.5. Characterization of synthesized copolymers – should be improve.

- 3.6. Encapsulation efficiency and drug loading – should be improve.

- Main findings should also be provided in conclusions.

- Make all references in same format for volume number, page numbers and journal name, because it is difficult to searching and reading.

- Furthermore, they should add the Graphical Abstract and polish manuscript using a high-quality native editing.

Based on these, I advise the authors to rectify the above mentioned errors and we hope to re-evaluate the revised manuscript.

Minor editing of English language required

Author Response

Comments:

This work concerns the Design, synthesis, and comparison of PLA-PEG-PLA and PEG-PLA-PEG copolymers for curcumin delivery to cancer cells. Important points are missing and there are some points that should be revised or corrected. Some important points are mentioned hereafter.

- In the Abstract, the authors need to improve with more specific short results and conclusions, i.e., academic novelty or technical advantages.

Response: We thank the reviewer for their comments. We made changes based on the opinion of the respected reviewer. (shown specifically in Page 1, line 27-29 and line 34-40).

- Keywords: add more keywords

Response: Following the suggestion of the reviewer, we added several new keywords (page 1, line 48-49).

- In the introduction, the authors do not show the significance and novelty of the work.

Response: We are grateful to the reviewer for providing this valuable feedback. Based on their insightful comments, we have added further description (Page 2, line 92-95).

- Give more detail for the Evaluation of 2.1. Force field.

Response: We appreciate the reviewer's valuable feedback, which helped us to improve the manuscript significantly. Based on their insightful comments, we have improved this section (Page 3, line 104-112).

- CH3CHOHCOOH……..to…………. CH3CHOHCOOH.

Response: We thank the Reviewer for careful consideration and have made the correction (page 4, Table. 1).

- HOCH2CH2OH……to………….. HOCH2CH2OH.

Response: We thank the Reviewer for their careful review, and made the correction (page 4, Table. 1).

- Please provides the references for equations and formula.

Response: We would like to express our gratitude to the reviewer for their careful consideration and valuable feedback. We have incorporated references for equations and formula (page 4).

- In part SEM: how the energy of the accelerator beam used?

Response: We would like to express our gratitude to the esteemed reviewer for their close attention and valuable insights in this section. In SEM, the electron beam was used as the primary beam to scan the sample surface. The energy of the electron beam was controlled using an accelerator, which accelerated the electrons to a specific energy level before they were focused onto the sample's surface.

- 3.5. Characterization of synthesized copolymers – should be improve.

Response: We thank the Reviewer for this helpful comment. As suggested, this section was modified and rewritten (Page 14, line 442-456).

- 3.6. Encapsulation efficiency and drug loading – should be improve.

Response: We appreciate the reviewer's helpful comment and have acted on their suggestion by modifying this section accordingly with a new paragraph (Page 16, line 487-494).

- Main findings should also be provided in conclusions.

Response: We are grateful to the reviewer for their helpful comment, and we have made some changes to the conclusion to focus on the key findings (Page 21, line 646-660).

- Make all references in same format for volume number, page numbers and journal name, because it is difficult to searching and reading.

Response: Thank you for your valuable comment. The reference section has been edited accordingly.

- Furthermore, they should add the Graphical Abstract and polish manuscript using a high-quality native editing.

Response: We appreciate the reviewer's helpful comment, and we have made efforts to eliminate linguistic and grammatical errors in our work.

Based on these, I advise the authors to rectify the above-mentioned errors and we hope to re-evaluate the revised manuscript.

Reviewer 2 Report

This study provides detailed information on the synthesis and characterization of the copolymers, as well as their in vitro release behavior and cytotoxicity against breast cancer cells. The authors also discuss the advantages of using copolymers in drug delivery systems and their potential applications. One major shortage of this study is no in vivo characterization of the copolymer (release and anti-cancer effects).

Overall, this study presents quite an extensive in-vitro analysis of the block copolymers PLA-PEG-PLA and PEG-PLA-PEG using various techniques, which is commendable. However, the following feedback may improve the paper:

1. Please provide a brief explanation of the significance and relevance of each technique (RMSD, RMSF, RDF, SASA, contact number analysis, distance analysis, MSD, energy calculations) at the beginning of the relevant subsection. This will help non-specialist readers understand the rationale behind choosing these techniques and how they contribute to the overall objective of the study.

2. In Section 3.1, you mention that "there was no significant difference in RMSD values between the two block copolymers, and the conformations of copolymers have not changed noticeably." Can you elaborate more on what implications these findings might have on your overall results or conclusions?

3. Please provide a little more context or explanation about the impact of the results in Section 3.2. Specifically, what does it mean that "a smaller distance and a larger g (r) indicated more penetration of water into the membrane"?

4. In Section 3.3, consider describing more about the impact of the differences in transverse and lateral diffusion on the behavior or potential applications of these block copolymers.

5. The results in Section 3.4 regarding energy calculations are intriguing. However, the terms used (Van der Waal energy, Electrostatic energy, Polar solvation energy, SASA energy, Entropy) could be briefly defined or explained to provide more clarity for readers unfamiliar with these terms.

6. The structure of the manuscript could be improved by adding a short summary of the main findings at the end of each subsection. This would help the reader to track the flow of results more easily.

7. Some sentences are quite long, consider breaking them into smaller ones for better readability.

Author Response

This study provides detailed information on the synthesis and characterization of the copolymers, as well as their in vitro release behavior and cytotoxicity against breast cancer cells. The authors also discuss the advantages of using copolymers in drug delivery systems and their potential applications. One major shortage of this study is no in vivo characterization of the copolymer (release and anti-cancer effects). Overall, this study presents quite an extensive in-vitro analysis of the block copolymers PLA-PEG-PLA and PEG-PLA-PEG using various techniques, which is commendable. However, the following feedback may improve the paper:

  1. Please provide a brief explanation of the significance and relevance of each technique (RMSD, RMSF, RDF, SASA, contact number analysis, distance analysis, MSD, energy calculations) at the beginning of the relevant subsection. This will help non-specialist readers understand the rationale behind choosing these techniques and how they contribute to the overall objective of the study.

Response: We thank the reviewer for letting us know. We have taken note of the reviewer's suggestion and have rechecked the manuscript accordingly.

  1. In Section 3.1, you mention that "there was no significant difference in RMSD values between the two block copolymers, and the conformations of copolymers have not changed noticeably." Can you elaborate more on what implications these findings might have on your overall results or conclusions?

Response: We are extremely grateful for the valuable feedback and attention provided by the esteemed reviewer. In molecular dynamics (MD) simulations, the RMSD (Root Mean Square Deviation) is often used as a measure of how far a simulated structure has deviated from an equilibrated structure. During an MD simulation, the system evolves over time towards equilibrium, where the average properties of the system no longer change significantly. The equilibration process can take some time, particularly for complex systems or when starting from an unphysical configuration. Once equilibrium is reached, the RMSD between subsequent frames of the simulation should remain relatively stable, indicating that the system has reached a stable state. However, during the equilibration phase, the RMSD will typically fluctuate widely as the system adjusts to the simulation conditions. Therefore, in practice, the RMSD is often used as a diagnostic tool to assess whether a system has equilibrated, and to monitor the stability of the simulation over time. A stable simulation with a constant RMSD typically indicates that the system has reached equilibrium and that the simulation is well-behaved.

Please provide a little more context or explanation about the impact of the results in Section 3.2. Specifically, what does it mean that "a smaller distance and a larger g (r) indicated more penetration of water into the membrane"?

Response: We wish to express our sincere gratitude to the esteemed reviewer for their valuable feedback and close attention to our work. In molecular dynamics simulations, the radial distribution function (g(r)) and distance analysis can be used to investigate the penetration of water molecules into lipid membranes. The radial distribution function represents the probability of finding a water molecule at a certain distance from a reference molecule, such as a lipid headgroup, while distance analysis tracks the distance between water molecules and the membrane surface over the course of the simulation. A smaller distance and a larger g(r) indicated more penetration of water into the membrane, it means that the water molecules are more likely to be found at smaller distances from the lipid headgroups and that there is a higher probability of water molecules being located closer to the membrane surface (Page 11, line 382-389).

  1. In Section 3.3, consider describing more about the impact of the differences in transverse and lateral diffusion on the behavior or potential applications of these block copolymers.

Response: We thank the Reviewer for careful consideration, and rewrote parts of the section some changes accordingly (Page 11, line 393-396, Page 20, line 602-610).

  1. The results in Section 3.4 regarding energy calculations are intriguing. However, the terms used (Van der Waal energy, Electrostatic energy, Polar solvation energy, SASA energy, Entropy) could be briefly defined or explained to provide more clarity for readers unfamiliar with these terms. it’s done

Response: We thank the reviewer for their close reading. We appreciate the reviewer's precise comments, and we have addressed all of their concerns accordingly (please look at section 2.4, in which we have made a number of changes and explanations).

  1. The structure of the manuscript could be improved by adding a short summary of the main findings at the end of each subsection. This would help the reader to track the flow of results more easily.

Response: We greatly appreciate the reviewer's precise comments and insights, and we have taken them into consideration while revising the manuscript. We have attempted to provide more concise explanations where possible and have also addressed the results more succinctly in the discussion section.

  1. Some sentences are quite long, consider breaking them into smaller ones for better readability.

Response: We appreciate the reviewer's attention and have made efforts to modify the sentences in line with their suggestions. If there is anything else we can do to improve the quality of our work, please do not hesitate to let us know.

Round 2

Reviewer 1 Report

The authors revised the manuscript according to the reviewers' comments.

Minor editing of English language required

Reviewer 2 Report

The authors have addressed all my comments. Thanks!